# Seed Size, Seed Dispersal Traits, and Plant Dispersion Patterns for Native and Introduced Grassland Plants

**DOI:** 10.3390/plants12051032

**Published:** 2023-02-24

**Authors:** Jane E. Tuthill, Yvette K. Ortega, Dean E. Pearson

**Affiliations:** 1Division of Biological Sciences, Ecology and Evolution, The University of Montana, Missoula, MT 59801, USA; 2Rocky Mountain Research Station, United States Forest Service, Missoula, MT 59801, USA

**Keywords:** dispersal adaptations, seed mass, seed traits, syndromes, trade-offs, trait databases

## Abstract

Most terrestrial plants disperse by seeds, yet the relationship between seed mass, seed dispersal traits, and plant dispersion is poorly understood. We quantified seed traits for 48 species of native and introduced plants from the grasslands of western Montana, USA, to investigate the relationships between seed traits and plant dispersion patterns. Additionally, because the linkage between dispersal traits and dispersion patterns might be stronger for actively dispersing species, we compared these patterns between native and introduced plants. Finally, we evaluated the efficacy of trait databases versus locally collected data for examining these questions. We found that seed mass correlated positively with the presence of dispersal adaptations such as pappi and awns, but only for introduced plants, for which larger-seeded species were four times as likely to exhibit dispersal adaptations as smaller-seeded species. This finding suggests that introduced plants with larger seeds may require dispersal adaptations to overcome seed mass limitations and invasion barriers. Notably, larger-seeded exotics also tended to be more widely distributed than their smaller-seeded counterparts, again a pattern that was not apparent for native taxa. These results suggest that the effects of seed traits on plant distribution patterns for expanding populations may be obscured for long-established species by other ecological filters (e.g., competition). Finally, seed masses from databases differed from locally collected data for 77% of the study species. Yet, database seed masses correlated with local estimates and generated similar results. Nonetheless, average seed masses differed up to 500-fold between data sources, suggesting that local data provides more valid results for community-level questions.

## 1. Introduction

The great variation in plant distribution patterns evident in nature derives in part from the differential ability of plant taxa to expand their populations into new areas. As sessile organisms, plants accomplish this primarily through seed dispersal. From a theoretical perspective, the Janzen–Connell hypothesis postulates that having seeds fall near the parent plant can have negative impacts on plant fitness because enemies drawn to parents can cause mortality to nearby seeds and seedlings [1,2], a pattern that has been supported in some systems ([3,4], but see [5]). More generally, many factors may drive selection for dispersal in plants, including reduced parent–offspring and sibling competition, inbreeding avoidance, and increased likelihood for species’ persistence in the face of environmental vagaries [6,7]. Hence, natural selection should favor dispersal adaptations that move seeds away from parent plants.

One seed trait that fundamentally affects seed movement is seed mass. Applying Newton’s second law of motion, the energy required to move a seed will increase as seed mass increases. Hence, all else being equal, larger seeds face greater dispersal challenges. Of course, all else is not equal, and every object experiences resistance to movement (counter forces) determined by the nature of specific environmental conditions in accordance with Newton’s third law of motion. Therefore, other seed attributes such as seed shape, including seed appendages, will influence dispersal in a manner determined by the physics of the dispersal forces encountered, e.g., air, water, and soil. Importantly, seed size is also embroiled in a critical life history tradeoff linked to the economics of reproduction. Within this tradeoff, resource constraints force parent plants to either produce a few large or many small seeds [8,9,10]. Plant species that produce many small seeds have a numerical fitness advantage, but individual seeds have minimal resources and a reduced survival probability, requiring that they reach favorable microsites [9]. In contrast, species producing few large seeds have higher per capita offspring fitness because larger seeds have more resources to survive stressful conditions and competition [9,11,12]. This seed-size fitness tradeoff is inextricably linked to the seed-size dispersal tradeoff because larger seeds require both more energy to produce and more energy to disperse. A key prediction from these combined theories is that species investing in larger seed size must also invest in dispersal to overcome dispersal constraints imposed by greater seed mass.

Global-scale tests of dispersal trade-offs have found that larger-seed sizes were associated with greater dispersal distances overall ([7,13] but see [14])—a finding attributed to a link between larger seed size and greater plant height [14,15]. However, seed mass is negatively related to dispersal distance across plant taxa when plant height is accounted for [14,16]. In combination, these findings support the hypothesis that larger seeds face greater dispersal constraints that must be overcome via plant adaptations such as height, but how does seed mass relate to seed adaptations for dispersal? In pine (*Pinus* spp.), samara size (an adaptation for wind dispersal) increased with seed mass up to a point after which wind adaptations declined and the seed dispersal mechanism shifted to zoochory [17]. Hence, selective pressures may shift dispersal away from abiotic forces and toward vertebrate seed dispersers at larger seed sizes see also [18,19,20,21]. Seed mass generally predicted dispersal adaptations across five temperate floras on three continents, with the smallest seeds lacking dispersal mechanisms and the largest seeds relying on vertebrate dispersal, whereas seeds in between favored wind at smaller sizes and invertebrate or vertebrate ectozoochory at larger sizes [21]. These patterns generally support the idea that larger-seeded plants that invest more energy in seed provisioning must also invest more energy in seed dispersal, but these relationships have not been well studied in grassland systems.

Given the fundamental importance of seed dispersal to plant ecology, it might be expected that seed dispersal strategies would be linked to plant dispersion patterns, especially for species actively expanding into new regions. In particular, one might expect this linkage to be stronger for newly introduced species as opposed to long-established natives because the role of dispersal in community assembly processes should be strongest at initial invasion stages and likely increasingly overshadowed by other processes as assembly progresses [22,23]. A correlation between seed dispersal traits, such as investment in seed dispersal appendages, and large geographical range (dispersal over 100 m in less than 50 years, according to [24]) has generally been found to be more prevalent in introduced species versus natives [25,26], although there are exceptions [27]. Regarding seed size, invasive species have larger seeds than native species in some systems [28,29], whereas in others, exotics are generally smaller than natives [30,31,32,33]. Overall, the relationship between seed dispersal traits and plant distribution patterns, particularly as it relates to provenance, remains unclear.

The growing availability of ecological databases has given rise to a pattern wherein trait-based ecology is increasingly relying on databases to quantify traits for testing ecological theory, rather than collecting data directly from the systems being studied [7,34,35,36]. While access to extensive datasets can be a powerful tool for addressing large-scale questions, this approach can lead to inaccurate study results, such as false positives in some cases [37]. The validity of such databases for different types of research questions remains largely unexplored, particularly for testing theory at community scales where locally collected data may better reflect the importance of community context (*sensu* [38]).

In this study, we quantified seed traits for 23 native and 25 exotic plant species found in bluebunch wheatgrass (*Pseudoroegneria spicata*) grasslands in western Montana, USA, to evaluate the relationship between seed size and seed dispersal traits and to evaluate the relationship between these traits and plant distribution patterns based on 620 1 m vegetation plots from 31 grassland sites. Our specific objectives were to: (1) quantify the relationship between seed mass and seed dispersal traits; (2) determine whether seed mass and other seed traits linked to dispersal correlate with plant dispersion patterns; (3) test whether these patterns are stronger for actively dispersing introduced species vs. long-established natives; and (4) examine the relationship between empirically collected seed trait metrics and those obtained from databases to assess the efficacy of databases for evaluating community-based questions.

## 2. Results

Introduced grassland species had significantly lower seed mass ($\bar{x}$ = 0.31 ± 0.11 mg) than native species ($\bar{x}$ = 1.07 ± 0.4 mg; *F*_1,46_ = 6.2, *p* = 0.017), while the proportion of species with seed dispersal structures did not differ by origin overall (native: 48% of n = 23, introduced: 36% of n = 25; *p* = 0.56). However, further analysis indicated linkages among all three factors. Although the presence of seed dispersal structures tended to vary with seed mass (*F*_1,44_ = 3.7, *p* = 0.063), this relationship differed by species origin (seed mass x origin: *F*_1,44_ = 5.7, *p* = 0.032), with a positive correlation for introduced taxa (Figure 1A). In sum, 83% of exotics with large seeds (>1.5 mg) had dispersal structures compared to only 21% of smaller-seeded exotics (*p* = 0.012), while the proportion of native species with these structures did not differ between the large-seed (45%) and small-seed (50%) classes (*p* = 1.0; Figure 1B). Origin significantly affected the presence of dispersal structures (*F*_1,44_ = 4.7, *p* = 0.036), but the pattern varied with seed mass as indicated by the significant interaction between origin and seed mass (test statistics above); introduced species were generally more likely than natives to have seed dispersal structures at large but not small seed masses (Figure 1).

Local-scale dispersion (proportion of plots occupied per site) correlated positively with seed mass for introduced but not native taxa, although this difference was only marginally significant (seed mass × origin: *p* = 0.067; Figure 2A, Appendix A). The seed mass relationship did not vary significantly with origin when we considered broad-scale dispersion (proportion of sites occupied; *p* = 0.16; Figure 2B, Appendix A). Neither dispersion metric was associated with the presence of dispersal structures (*p* > 0.12; Appendix A).

Empirically estimated seed masses from local grasslands generally differed from those obtained from the TRY trait database. Of the 44 species tested, seed masses differed significantly (*p* < 0.05) in 77% of cases, with mean seed masses diverging more than 2-fold for 21% of species and by as much as 500-fold in one case (*Lomatium triternatum*; Figure 3, Appendix A). Despite this result, seed masses from the two sources were highly correlated (*r*^2^ = 0.7, *p* < 0.0001), and in most cases, rank order did not shift dramatically due to the great range of variation in seed size across species (Figure 3, Appendix A). Accordingly, seed mass relationships were qualitatively similar when we used estimates derived from TRY instead of measures from the current study to conduct the analyses described above (Appendix A).

## 3. Discussion

Plants rely primarily on seed dispersal to expand their populations, yet the linkage between seed traits, plant movement, and ultimately plant distribution patterns is unclear. To explore this question, we quantified seed traits for 48 species of native and introduced grassland plants and related seed traits to plant distribution. We found that larger-seeded species were more likely to have dispersal adaptations such as pappi and awns, but this pattern was only evident for exotics. Dispersal adaptations may be important for larger-seeded invaders to overcome the combined constraints of seed mass and dispersal barriers to invasion. These larger-seeded exotics also tended to be more widely distributed than their smaller-seeded counterparts, again a pattern that was not apparent for native taxa. Our results suggest that the role that seed traits play in affecting plant dispersal and distribution patterns for actively expanding populations may be obscured for long-established populations by other ecological filters such as competition or herbivory. In comparing seed masses estimated from locally collected data to those in the TRY traits database, we found that estimates differed significantly for most species, with some diverging by as much as 500-fold. While these differences did not affect our qualitative conclusions in this study, such divergence from local community trait estimates could confound many local-scale questions. Below, we expound on the implications of these results.

Our finding that introduced species with greater seed masses tended to also have dispersal adaptations such as awns and pappi suggests that some species that invest more energy in seed provisioning vs. seed number are capable of also investing more energy in seed dispersal. In the context of the seed size-fecundity tradeoff [9,10], this result implies that some large-seeded species are not expending all available reproductive resources on seed size alone but are also allocating some of that energy toward the dispersal of those seeds to overcome constraints of increased mass on dispersal. As such, this result not only underscores a linkage between the seed-size fecundity tradeoff and the seed-mass dispersal tradeoff, but it also shows that some species may balance these tradeoffs to increase provisioning while still moving larger seeds. We also found that exotics had smaller seeds overall than the natives. Overall, exotic seed masses exhibited a bimodal distribution wherein they had either very small seeds compared with most native species or large seeds that were also endowed with dispersal adaptations. Collectively, these results suggest that exotics in our system may be exploiting two divergent strategies not well represented by native taxa, fitting with the idea that some invaders gain access to communities by exhibiting strategies underrepresented among native species [33,39,40,41,42].

In theory, the role of dispersal should be strongest during the initial stages of community assembly, with biotic interactions increasingly overshadowing these effects as assembly progresses due to the hierarchical nature of assembly processes [22,23]. Our finding that the correlation between seed traits and plant dispersion patterns could only be demonstrated for actively expanding introduced species and not for long-established natives is consistent with this idea. Seed traits may help to predict plant dispersion at least early in the invasion process, but over time processes such as competition, herbivory, etc., may obscure such effects. Despite these hierarchical processes masking initial dispersal effects, it is important to note that species can only persist where they arrive and establish. Dispersal effects may be difficult to track over time, but this does not undermine the importance of the early filtering effects that seed traits may have on plant movement and establishment [16,43,44]. Many studies have shown that dispersal traits, such as seed dispersal appendages, are more expressed in introduced species than natives [25,26], but others have shown the opposite [27]. One study found that greater seed size within a species may be linked to greater invasibility at the invasion front [45]. It is likely that multiple traits and multiple filters shape invasion patterns in plant species, including not only seed size [46] but also growth form [47], seed longevity [48], and propagule pressure [49], thereby confounding the independent effects of individual processes.

As some studies in trait-based ecology are increasingly relying on databases to quantify traits for testing ecological theory [7,34,35,36], the opportunities for inaccurate study results are also rising and have been discovered in some cases [37,50]. That the empirically collected seed masses in this study differed significantly from those obtained from the TRY trait database for 77% of our study species is a concerning result. Seed mass can vary greatly due to soil resources, precipitation, and temperature [51,52], and therefore, the greater the geographic and ecological variation among communities sampled, the more seed mass measures may diverge. Indeed, the majority of our study species occur on multiple continents. Certainly, methodological variables such as whether or not dispersal structures are included and the degree to which seed quality is accounted for will also influence seed mass measures, and such criteria may not follow the data, especially as they are subsumed into larger and larger collective databases. Indicative of the latter process, we found that 55% of the seed mass records we obtained from the TRY trait database represented duplicate values, an estimate we deem conservative (see Materials and Methods for details). Errors or inconsistencies in the data will of course be amplified as data are duplicated, and accurate duplicates would falsely reduce variance. Nonetheless, when we used the TRY database seed masses to revisit the analyses conducted here, this did not qualitatively change any of our conclusions. This outcome arose from the fact that the range of seed sizes was so large among our study species that the relative ranking of species seed masses did not shift much by data sources. Hence, for broad, cross-species sorts of analyses, the deviation between the database and empirically measured data may not confound conclusions. However, caution should be taken when using database information, particularly for local scale quantitative analyses where local trait measures from the system should best reflect true trait values at that scale.

Our results suggest that seed traits may influence plant distribution patterns via their effects on seed dispersal. However, the legacy of such effects may be quickly overshadowed (in ecological time) by other processes such as competition, herbivory, and abiotic conditions. Hence, the role that seed traits play in driving plant distribution patterns may be most readily observed among newly expanding species such as introduced plants, and seed traits may play important roles in determining invasion status. Additional studies that explore the relationships between seed traits and plant distribution patterns for native and introduced species could expand our understanding of community assembly and plant distribution patterns.

## 4. Materials and Methods

Our study took place in the semi-arid grasslands of the Intermountain Region in western Montana, U.S.A. The native system is dominated primarily by bluebunch wheatgrass (*Pseudoroegneria spicata*), with other grasses and a great variety of forbs diversifying the system [53], but it is heavily invaded by exotics [54]. We identified our study species, comprised of 23 native and 25 exotic species (Appendix A), to reflect a range of dispersion patterns by using data from 620 1-m^2^ vegetation plots from 31 grassland sites spread over 20,000 km^2^ of this region (see [54] for details). Plant dispersion patterns were defined at a local scale by the proportion of plots occupied within a site and at a broad scale by the proportion of sites occupied per species. For each species, we collected at least 50 seeds from each of the 10 plants at each of 3 locations in Missoula and Lake County, Montana, in either 2020 or 2021. Collection locations were chosen opportunistically based on species presence and hence differed by species. Although these locations did not align with sites surveyed for species dispersion per se, they were generally drawn from the central portion of the study area. Seeds were stored in a laboratory under ambient conditions until measurements were taken, at which point they were cleaned by hand and sorted based primarily on visual characteristics to remove potentially non-viable seeds. To determine the mean seed mass per species, we weighed a fixed number of samples (three or four) from each of the three locations. The number of seeds weighed per sample was set per species to ensure a total mass > 1.5 mg, the minimum reading needed for an accuracy of 2% per the specifications of the balance. For 32 of our 48 species, only 10 seeds were needed to reach this minimum. For the remaining species, we increased the number of seeds included per sample in increments of 10 (range 20–150 seeds/sample) until the minimum mass was reached. Seed mass included the entire diaspore (e.g., endosperm, seed coat, awns, and dispersal appendages) to ensure that all species could be treated in the same way (e.g., dispersal appendages such as wings would have been very difficult to remove from small-seeded species). Though the inclusion of dispersal appendages potentially biases seed mass estimates for this subset of species, we note that this bias should be small relative to the large variation in seed mass across species. Indeed, estimates for three exotic species (*Lactuca serriola*, *Taraxacum officinale*, and *Tragopogon dubius*) with pappuses showed that these structures increased seed mass measures by <12%.

For the remaining measurements, we used a ProgRes C10 camera (Jenoptik, CCD/CMOS) to create images of 20 seeds per species drawn from the 3 sampling locations (n = 6 from two locations and n = 8 from the third, chosen randomly). We used the images to obtain the following measurements for each seed via ImageJ software [55]: seed length (maximum), seed width (maximum), and seed surface area. These seed measurements excluded dispersal structures. We inspected seeds to record whether seeds of each species possessed dispersal structures, including pappuses, awns, wings, or plumes. For smaller-seeded species, we accomplished this using seed images and also checked the literature to assure that dispersal structures were not missed.

To enable comparison of empirical seed measures to those available in online trait databases, we used the TRY plant trait database (accessed 22 September–7 October 2022), a global database integrating ~700 datasets, including other major collective databases [56]. This database included seed mass data for 44 of our 48 species but contained insufficient data to evaluate the other seed traits (i.e., length, width, and surface area) we measured (i.e., for only 2–40% of our study species). Importantly, 63% of n = 831 seed mass records obtained from the TRY database could not be used in analyses. This is because these contained duplicate data that resulted from the consolidation of many datasets from common sources, particularly the KEW Seed Information Database [57] and the LEDA Traitbase [58]. Specifically, 48% of the records we obtained were marked by TRY as duplicates, and we identified another 7% as probable duplicates, usually because they matched other records identified by TRY as duplicates (86% of n = 59 cases). In the remaining cases, we identified them as probable duplicates because records matched other values and involved at least one of the major collective databases (i.e., KEW or LEDA). While we suspect that our method of screening for duplicates was conservative, we otherwise had no efficient way to determine whether similar values (which were identical when rounded) were derived from the same original source. We also discounted another 8% of seed mass records obtained from the TRY database because they were marked as minimum or maximum values, which repeated information represented by standard, mean values.

### Statistical Analysis

We used SAS version 9.4 [59] to examine relationships between species seed size, seed dispersal traits, and dispersion. Given that seed mass was highly correlated with seed length (Spearman *r*^2^ = 0.76, *p* < 0.0001), width (Spearman *r*^2^ = 0.73, *p* < 0.0001), and surface area (Spearman *r*^2^ = 0.84, *p* < 0.0001), we focused analyses on seed mass, in alignment with previous studies of seed size patterns and the availability of data online, e.g., in the TRY database. The mean values per species for all seed size measurements are included in Appendix A, and the mean seed mass on the log scale (natural log of the raw value plus one) was used to represent each species in analyses.

To examine the relationship between seed dispersal traits and seed mass as a function of species origin, we treated the presence of seed dispersal structures as the response in a generalized linear model (GLM) with a binary distribution; seed mass, species origin (native or introduced), and their interaction were included as dependent variables. We also compared the proportion of species with seed dispersal structures between “small” (<1.5 mg) and “large” (>1.5 mg) seed size classes for native and exotic species, respectively, via Fisher exact tests. Seed size classes were defined based on a natural break in the seed mass data (Appendix A), which also split native species as close as possible to the median value (1.3 mg).

We related plant dispersion metrics (local scale: mean proportion of plots occupied per site, broad scale: proportion of sites occupied) to seed traits (seed mass, presence of seed dispersal structures) and species origin using GLMs with a beta distribution. Separate models were constructed with each dispersion metric as the response and each seed trait as a dependent variable along with the origin and the seed trait × origin interaction.

To compare seed mass between our measurements and the values obtained from the TRY database for each species, we used one-sample *t*-tests. This approach allowed us to compare mean values between data sources without accounting for the variance associated with the TRY estimate, which was derived from only one value for 11% of species and from less than five values for 48% of species (sample size range = 1–30, $\bar{x}\;$= 7.0 ± 1.0; Appendix A).

## Figures and Tables

**Figure 1 plants-12-01032-f001:**
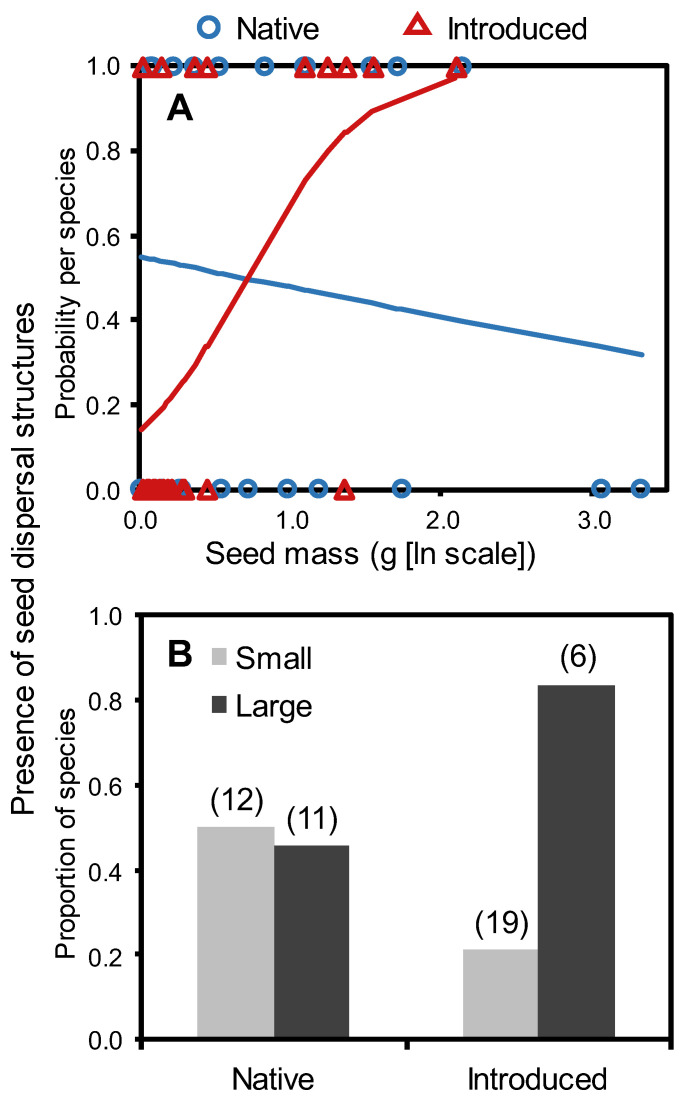
Relationship between seed mass (ln [mg + 1]) and the presence of seed dispersal structures, as compared between 23 native and 25 introduced grassland species in Montana, USA, in two analyses. (**A**) The dispersal structure-seed mass relationship varied significantly by species origin when all species were tested together (see text), with introduced species showing a positive relationship and native species trending in the opposite direction, as represented by solid lines indicating predicted probabilities. Note that the relationship was significant for introduced species when tested alone (*F*_1,23_ = 5.6, *p* = 0.027), but not for native species (*F*_1,21_ = 0.4, *p* = 0.56). Open symbols represent observed values (1 = species with seed dispersal structure, 0 = without). (**B**) The proportion of species with dispersal structures was four times higher for large (>1.5 mg [ln scale: >0.9 mg]) vs. small-seeded exotics but did not differ between seed mass categories for native species (see text). Sample sizes (number of species) are shown above bars.

**Figure 2 plants-12-01032-f002:**
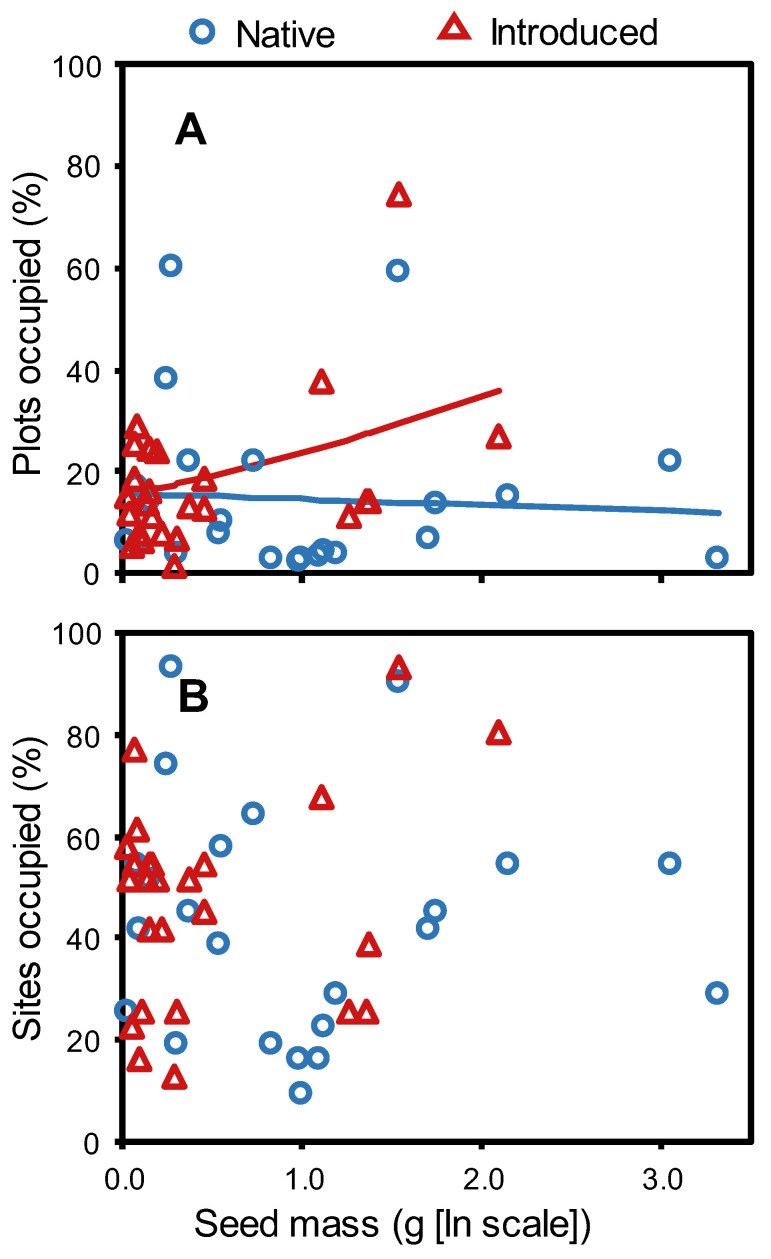
Relationship between seed mass (ln [mg + 1]) and species dispersion measured at two scales, as compared between 23 native and 25 introduced grassland species in Montana, USA. (**A**) At the local scale (proportion of plots occupied per site), the dispersion-seed mass relationship varied marginally by species origin when all species were tested together (*p* = 0.067), with introduced species showing a positive relationship and native species trending in the opposite direction, as represented by solid lines indicating predicted probabilities. Note that the relationship was significant for introduced species when tested alone (*F*_1,23_ = 5.6, *p* = 0.027), but not for native species (*F*_1,21_ = 0.2, *p* = 0.69). (**B**) At the broad scale (proportion of sites occupied), dispersion did not vary with seed mass for either native or introduced species, whether tested together (see text) or separately by origin (native: *F*_1,21_ = 0.1, *p* = 0.78; introduced: *F*_1,23_ = 2.9, *p* = 0.1).

**Figure 3 plants-12-01032-f003:**
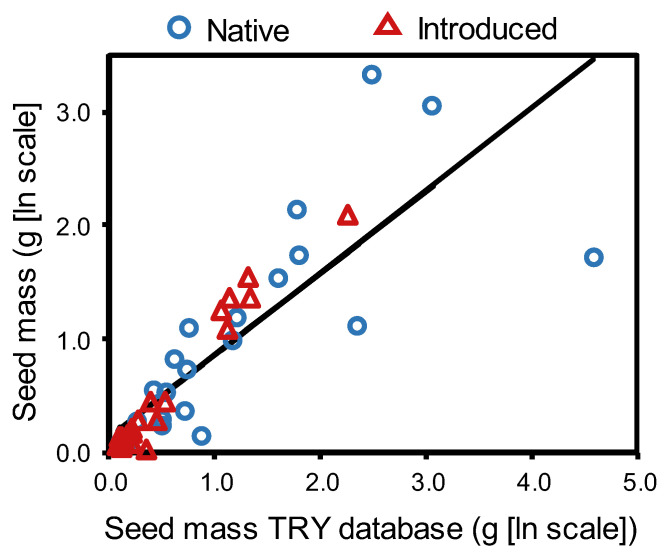
Seed mass (ln [mg + 1]) as compared between local measurements and values obtained from the TRY trait database for 23 native and 25 introduced grassland species in Montana, USA. Despite large variation between data sources for some species, seed mass measures were significantly correlated (see text); solid line represents the predicted relationship, and open symbols show observed values.

## Data Availability

Data is archived in Dryad. https://doi.org/10.5061/dryad.2z34tmpr2.

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
