# Peer review of "Seed Size, Seed Dispersal Traits, and Plant Dispersion Patterns for Native and Introduced Grassland Plants"

_plants, 2023, doi:10.3390/plants12051032_

Round 1

Reviewer 1 Report

This manuscript, “Seed size, seed dispersal traits, and plant dispersion patterns for native and introduced grassland plants” by Tuttle et al, is a much needed contribution to the field of native seed ecology and trait-based science, which needs more data driven research to ground theory. This is a very creative study, considering seed characteristics (size and dispersal appendages) for a wide array of native and introduced species in a Montana grassland system, asking about the relationship between seed size and the presence of dispersal appendages, along with the relationship between seed size and current distribution. They also include a compelling comparison of the seed mass information in the TRY database with the information from their field studies.

The authors find differences in native and introduced species, specifically the link between seed mass and dispersal appendages exist in the introduced species, but not in native ones. They also find that the TRY database has limitations and does not match their field data, but interestingly, a strong correlation between field and reported data that means that the outcomes of a broad question with many species would be the same, with field or TRY data. 

I suggest several changes below, including more clarity in the methods, considering adding a 3rd “medium” category or eliminating that analysis, and potentially the addition of another figure. Specific suggestions below.

Abstract

Below, under methods and discussion, I suggest some more nuance about “large” seeds in the introduced population (basically, there weren’t any, only small and medium), and suggest that the abstract change accordingly to reflect this.

Introduction

I really appreciated that the introduction provides a concise yet detailed overview of seed trait ecology. The authors focus mostly on dispersal and may benefit from mentioning other trade-offs that may be occurring with seed size (e.g. germination, dormancy, growth rate) all of which may influence invasion ecology.

The authors measured traits in grasslands yet use many tree examples in their introduction. Consider using examples more similar to the study system, or state that less work has been done in grassland systems to explain this.

Lines 63-67 seem contradictory. I think you are saying that the global pattern is an artifact? Perhaps changing “Notably” to “However” would make this clear.

Lines 66-72, 76-77: Suggest changing from “so and so found” to “factual information (cite so and so”) format. 

Line 96: You might be interested in this very fun paper that shows differences in seed mass in plants at the invasion edge (vs. center) of the population! 

Rice, K.J., Gerlach, J.D., Dyer, A.R. et al. Evolutionary ecology along invasion fronts of the annual grass Aegilops triuncialis . Biol Invasions 15, 2531–2545 (2013). https://doi.org/10.1007/s10530-013-0471-6

Lines 102-107: Nice point

Methods and results

Researchers weighed seeds with appendages on and then state that mass is correlated with presence of appendages. These appendages are quite small, but is there a possibility that weighing the seed all together is causing this relationship? Some of these seeds only weigh 0.02-0.07 mg, how much of that weight is the appendage? It would be nice to see an acknowledgement of this, even if researchers only weighed a few seeds with appendages removed (e.g. with and without awn).

Did you consider measuring the size or mass of the appendages? How were seeds cleaned? Was there a possible loss of smaller seeds (common on some equipment) Did authors consider viability of seeds before weighing?

The authors sampled seeds from 3 of the sites then associated that data with presence/absence for 620 plots across 20,000 km2. It is not clear where the sampled sites are within the broader sampling range. I’d like to see more details about sample locations, even if it's a supplemental table or figure. 

Lines 271-272 it is not clear how many total seeds were collected, was it 50 total seeds or 50 from each of 10 plants?

The fact that researchers weighed different numbers of seeds depending on weight to get an average was a little concerning. Can you provide an example here in your text (“....accuracy of 2% per the specifications of the balance. That is, if a seed weighed X, our balance measured to Y, so we would XYZ for this sample”), and give the range of seeds measured? Was it 10-100, 10-1000?

Line 282: “as supported by information from…” A bit awkward; what do you mean? Seems like the images aren’t necessary to record dispersal structures, simply looking at the seeds would do it? 

Line 291: clarify what traits you mean (“i.e., seed length, seed width, and surface area information was only available for 2-40% of….”)

Line 292: Instead of Foremost, say “This is because…” I had to read this several times to understand that you were explaining why you couldn’t use many of the seed mass records.

Lines 303-305: Clarify this; I read it several times and don’t understand what you mean by “and the associated mean value was also apparent in most cases” (why would you then throw out that 8%? I think I just don’t know what you are trying to say).

I appreciate the authors analyzing and presenting the data both as continuous seed mass and as categorical (it’s a little easier to see the patterns in the categories), however, I question the appropriateness of the two size categories. In particular, the introduced species simply don’t have large seeds; I see more of a small/medium (which would be the max of introduced species) and then large (which would only be native). I understand that this is not amenable to the same analysis, but as it is, it is a bit misleading to talk about small and large seeds for native and introduced species as if they are the same thing. They really are not. I would consider using more categories (at least three) or removing the categorical analysis all together. 

Finally, this study only looks at the presence or absence of dispersal appendages rather than type or size. In the supplementary doc they do differentiate dispersal structures. It's probably a low sample size issue, but appendages are not all the same, especially in terms of dispersal distances; that could be brought up in the discussion.

Figures and tables

Figure 1: Again, the “large” category is pretty misleading in panel B. I really think it should be small, medium, and large, with a missing panel for "large" for introduced species. I understand that this might not be analyzable but better to show it in a non-misleading way, without an analysis, than this.

Table S2: Consider including variance in tables in addition to means. The reason I suggest this is that I was wondering if findings from Montana overlap the Try database at all?

Discussion

Lines 184-186, which is: “We found that larger-seeded species were more likely to have dispersal adaptations such as pappi and awns, but this pattern was only evident for exotics.”

Here, and in the abstract, it seems not totally accurate to phrase your results this way, as it was more like… medium-seeded exotics were more likely to have dispersal adaptations such as pappi and awns. There were no larger-seeded exotics.

Line 190: affecting, not effecting

Line 208-210: “Overall, exotic seed mass exhibited a bimodal distribution wherein they had either very small seeds compared with most native species, or they had large seeds that were also endowed with dispersal adaptations.” 

This is a super cool result. I would argue that they have either very small seeds or MEDIUM seeds with dispersal adaptations. I think this should be in your results, in a figure (maybe instead of the categorical ones). I think it’s pretty interesting that none of the introduced species have the largest seeds; maybe the dispersal barriers are just too high for them to get here?

Line 219: Could be another place to talk about that goatgrass paper reference pasted above; talk about active expansion, the leading edge of an invasive patch!

Line 228: Suggest “Some” instead of “Many”

I’m not sure where you are on word limit, but consider ending on a more wrap-up note in a final paragraph, rather than a caveat sentence/paragraph.

Author Response

Reviewer 1

Comments and Suggestions for Authors

This manuscript, “Seed size, seed dispersal traits, and plant dispersion patterns for native and introduced grassland plants” by Tuttle et al, is a much needed contribution to the field of native seed ecology and trait-based science, which needs more data driven research to ground theory. This is a very creative study, considering seed characteristics (size and dispersal appendages) for a wide array of native and introduced species in a Montana grassland system, asking about the relationship between seed size and the presence of dispersal appendages, along with the relationship between seed size and current distribution. They also include a compelling comparison of the seed mass information in the TRY database with the information from their field studies.

The authors find differences in native and introduced species, specifically the link between seed mass and dispersal appendages exist in the introduced species, but not in native ones. They also find that the TRY database has limitations and does not match their field data, but interestingly, a strong correlation between field and reported data that means that the outcomes of a broad question with many species would be the same, with field or TRY data. 

I suggest several changes below, including more clarity in the methods, considering adding a 3rd “medium” category or eliminating that analysis, and potentially the addition of another figure. Specific suggestions below.

Abstract

Below, under methods and discussion, I suggest some more nuance about “large” seeds in the introduced population (basically, there weren’t any, only small and medium), and suggest that the abstract change accordingly to reflect this.

Response: In the Abstract, only the term “larger” not “large” is used.  This is a relative term indicating that the larger of the sampled seeds reflected these attributes.  It does not imply that the seeds themselves were large in an absolute sense.  Hence, we did not change this text.  We address the review’s general point about exotic seed sizes in greater detail below where the reviewer makes more detailed comments on this topic.

Introduction

I really appreciated that the introduction provides a concise yet detailed overview of seed trait ecology. The authors focus mostly on dispersal and may benefit from mentioning other trade-offs that may be occurring with seed size (e.g. germination, dormancy, growth rate) all of which may influence invasion ecology.

Response: We appreciate these comments.  We agree that germination, dormancy, growth rate could also be affected, but we did not add text to include these other traits because we did not examine them in this study.

The authors measured traits in grasslands yet use many tree examples in their introduction. Consider using examples more similar to the study system, or state that less work has been done in grassland systems to explain this.

Response: We did lean on literature from forested systems specifically because these relationships have been less well studied in grasslands.  We now state this limitation at the end of this paragraph in the Introduction, as suggested by the reviewer.

Lines 63-67 seem contradictory. I think you are saying that the global pattern is an artifact? Perhaps changing “Notably” to “However” would make this clear.

Response: we made this change as suggested.

Lines 66-72, 76-77: Suggest changing from “so and so found” to “factual information (cite so and so”) format. 

Response: We made these changes as suggested.

Line 96: You might be interested in this very fun paper that shows differences in seed mass in plants at the invasion edge (vs. center) of the population! 

Response: We appreciate the reviewer sharing this paper.  We have included it in the Discussion.

Rice, K.J., Gerlach, J.D., Dyer, A.R. et al. Evolutionary ecology along invasion fronts of the annual grass Aegilops triuncialis . Biol Invasions 15, 2531–2545 (2013). https://doi.org/10.1007/s10530-013-0471-6

Lines 102-107: Nice point

Methods and results

Researchers weighed seeds with appendages on and then state that mass is correlated with presence of appendages. These appendages are quite small, but is there a possibility that weighing the seed all together is causing this relationship? Some of these seeds only weigh 0.02-0.07 mg, how much of that weight is the appendage? It would be nice to see an acknowledgement of this, even if researchers only weighed a few seeds with appendages removed (e.g. with and without awn).

Response: We now make clear that we included seed dispersal structures in seed mass measures to ensure that all species could be treated the same, i.e., to account for the fact these structures would have been difficult to impossible to remove for some species. This was the case for species like Achillea millefolium and Boechera retrofracta, which have very small seeds (length <2 mm) with winged margins. Similarly, the pappuses of species like Centaurea stoebe do not readily detach, nor do the awns of most grasses. 

That said, we do have data for three exotic species to show that the mass of their pappuses was relatively small, which is not a surprise given the general tradeoff between mass and mobility.  Specifically, pappus mass represents 6.3% of the total mass per seed for Lactuca serriola, 11.9% for Taraxacum officinale, and 11.4% for Tragapogon dubius.  A much larger shift in seed mass would be needed to change assignment to seed mass categories for any species with dispersal structures, and T. dubius remains the largest-seeded exotic by far regardless of whether its pappus is discounted.  Furthermore, as now explained in the text, the shift in mass noted for these three exotics is small compared to the variation in seed mass seen across species. The influence of the latter was underscored when we repeated our analyses using seed mass estimates from TRY, which showed that observed seed mass relationships were robust to much larger shifts in seed mass values. 

Did you consider measuring the size or mass of the appendages? How were seeds cleaned? Was there a possible loss of smaller seeds (common on some equipment) Did authors consider viability of seeds before weighing?

Response: We have added text to the Methods to better clarify these points.  Seeds were carefully cleaned by hand.  Yes, viability was considered, and we attempted to use only viable seeds based primarily on visual characteristics.  We did measure awn length and attempted to measure surface area of pappuses but did not include analysis of these metrics given insufficient sample size for any one type of dispersal structure (n < 6 species per origin category) coupled with the difficulty of validly quantifying their size in the face of variable morphologies.

The authors sampled seeds from 3 of the sites then associated that data with presence/absence for 620 plots across 20,000 km2. It is not clear where the sampled sites are within the broader sampling range. I’d like to see more details about sample locations, even if it's a supplemental table or figure. 

Response: We have added more information about seed sampling locations to the text.  These locations were chosen opportunistically based on species presence and hence differed by species. Although collection locations did not align with sites surveyed for species dispersion per se, they were generally drawn from the central portion of the study area.

Lines 271-272 it is not clear how many total seeds were collected, was it 50 total seeds or 50 from each of 10 plants?

Response:  We now clarify that we collected a minimum of 50 seeds from each of 10 plants per location.

The fact that researchers weighed different numbers of seeds depending on weight to get an average was a little concerning. Can you provide an example here in your text (“....accuracy of 2% per the specifications of the balance. That is, if a seed weighed X, our balance measured to Y, so we would XYZ for this sample”), and give the range of seeds measured? Was it 10-100, 10-1000?

Response: While it would have been simpler to weigh the same number of seeds per sample to determine the mean mass per seed, this was impossible given the thresholds of the balance coupled with the great range in seed mass among species.  As now specified in the text, the minimum mass needed to assure an accuracy of 2% on the balance used was 1.5 mg.  This translated to 150 seeds for our smallest-seeded species.  However, 150 seeds of our largest seeded species would amount to >4000 mg, which far exceeds the measurement capability of our analytical balance.  Hence, we developed a process that met the bounds of the balance while being as efficient as possible.  We now elaborate on the process and include the range of seeds measured (n=10 for 32 of 48 species, n=20-150 for those remaining).

Line 282: “as supported by information from…” A bit awkward; what do you mean? Seems like the images aren’t necessary to record dispersal structures, simply looking at the seeds would do it? 

Response: We rephrased this sentence and now explain, “We inspected seeds to record whether seeds of each species possessed dispersal structures including pappuses, awns, wings, or plumes. For smaller-seeded species, we accomplished this using seed images and also checked the literature to assure that dispersal structures were not missed.”

Line 291: clarify what traits you mean (“i.e., seed length, seed width, and surface area information was only available for 2-40% of….”)

Response: We added this clarification as suggested.

Line 292: Instead of Foremost, say “This is because…” I had to read this several times to understand that you were explaining why you couldn’t use many of the seed mass records.

Response: We made this change as suggested.

Lines 303-305: Clarify this; I read it several times and don’t understand what you mean by “and the associated mean value was also apparent in most cases” (why would you then throw out that 8%? I think I just don’t know what you are trying to say).

Response: We now clarify that minimum and maximum values were excluded because they repeated information represented by the standard, mean values given in the TRY database.

I appreciate the authors analyzing and presenting the data both as continuous seed mass and as categorical (it’s a little easier to see the patterns in the categories), however, I question the appropriateness of the two size categories. In particular, the introduced species simply don’t have large seeds; I see more of a small/medium (which would be the max of introduced species) and then large (which would only be native). I understand that this is not amenable to the same analysis, but as it is, it is a bit misleading to talk about small and large seeds for native and introduced species as if they are the same thing. They really are not. I would consider using more categories (at least three) or removing the categorical analysis all together. 

Response: This assessment is not accurate.  The categorical break for “large” vs “small” seeds is 0.9 mg on the log scale (1.5 mg on the original scale).  As explained in the text, this represents a natural size break in the data, which shows a distinct cluster below and above this threshold (Table S2, Fig. 1). This break splits native species evenly and for exotics, spans an impressive seed mass range (0.5 – 1.1 on the log scale, 0.6 – 2.0 on the original scale) wherein no species fall. With this break, there are 6 large-seeded exotics and 11 large-seeded natives.  The seed sizes represented by the large-seeded exotics overlap the range represented by native species to a large extent, and there are only two native species with larger seeds than seen for exotics.  So, while it is true that there are fewer large-seeded exotics, as clearly shown in the statistical comparison of seed masses between these groups, there are in fact large-seeded exotics represented, and we feel that the number is reasonable given the strong size break evident. For these reasons, we have retained the data and figures are they are.  However, we note that this size break is clearly stated in the Fig. 1 caption so that readers can evaluate the data accordingly.

Finally, this study only looks at the presence or absence of dispersal appendages rather than type or size. In the supplementary doc they do differentiate dispersal structures. It's probably a low sample size issue, but appendages are not all the same, especially in terms of dispersal distances; that could be brought up in the discussion.

Response:  Even in a reasonable sample of 48 species, sample size quickly falls off to levels too small for analyses when the traits are broken into too many categories.  We had 9 species with awns (4 exotic, 5 native), 7 with pappi (5 exotic, 2 native), 3 with wings (all native), and 1 native with plumes.  Hence, there are just insufficient data in several categories to do much with these samples. 

Figures and tables

Figure 1: Again, the “large” category is pretty misleading in panel B. I really think it should be small, medium, and large, with a missing panel for "large" for introduced species. I understand that this might not be analyzable but better to show it in a non-misleading way, without an analysis, than this.

Response: Please see our explanation above.

Table S2: Consider including variance in tables in addition to means. The reason I suggest this is that I was wondering if findings from Montana overlap the Try database at all?

Response: We opted to include sample size but not variance in the table for efficiency and simplicity, as our goal was to squeeze in mean values for all seed size measures while also allowing easy visual comparison between data sources for seed mass.  As explained in the text, variance for TRY values could not be estimated in many cases given samples sizes as low as n=1.  For this reason, our statistical test for differences between TRY’s seed mass values and ours accounted only for the variance around the latter, as analyzed on the natural log scale.  Hence, in cases where p<0.05, the TRY value did not overlap the 95% interval defined by this variance, i.e., the test explicitly addresses this question.

Discussion

Lines 184-186, which is: “We found that larger-seeded species were more likely to have dispersal adaptations such as pappi and awns, but this pattern was only evident for exotics.”

Here, and in the abstract, it seems not totally accurate to phrase your results this way, as it was more like… medium-seeded exotics were more likely to have dispersal adaptations such as pappi and awns. There were no larger-seeded exotics.

Response:  Please see our response above.  There were in fact 6 large-seeded species compared with 11 large-seeded natives.

Line 190: affecting, not effecting

Response: We made this change as suggested.

Line 208-210: “Overall, exotic seed mass exhibited a bimodal distribution wherein they had either very small seeds compared with most native species, or they had large seeds that were also endowed with dispersal adaptations.” 

This is a super cool result. I would argue that they have either very small seeds or MEDIUM seeds with dispersal adaptations. I think this should be in your results, in a figure (maybe instead of the categorical ones). I think it’s pretty interesting that none of the introduced species have the largest seeds; maybe the dispersal barriers are just too high for them to get here?

Response: Please see our response above about the seed size breaks.  Even so, we agree with the reviewer that this tendency toward smaller exotics and moreover toward the largest-seeded species among the exotics having dispersal adaptations is quite interesting and worthy of further investigation in other grasslands and with larger sample sizes.

Line 219: Could be another place to talk about that goatgrass paper reference pasted above; talk about active expansion, the leading edge of an invasive patch!

Response: We have added a sentence addressing this paper at this point in the Discussion.

Line 228: Suggest “Some” instead of “Many”

Response: We changed many to some.

I’m not sure where you are on word limit, but consider ending on a more wrap-up note in a final paragraph, rather than a caveat sentence/paragraph.

Response: We have added a final concluding paragraph.

Reviewer 2 Report

In general, manuscript “Seed size, seed dispersal traits, and plant dispersion patterns for native and introduced grassland plants” (plants-2213896) by Jane E. Tuthill, Yvette K. Ortega, and Dean E. Pearson is very interesting and well written. Authors studied the relationship between size and seed dispersal traits in native and exotic grassland species in western Montana, USA in order to assess factors that influence the efficiency of active dispersal. In the face of biological invasions, I find this this subject to be important. Language is clear and smooth. However, I suggest some corrections and explanations in the text:

P. 1, L. 32-35: The construction of this sentence is strange. On the one hand, authors wrote: "From a theoretical perspective...", on the other hand, the papers tested this hypothesis were cited. Please, explain the point of hypothesis in the first sentence, and later, give suitable examples.

P. 2, L. 60-62: It is not always true. What about the seeds dispersed due to the gravity (Quercus, Fagus, Aesculus, Lagenaria, Cocos)? In so formulated sentence, height of maternal plant (from the next paragraph) cannot be considered as "dispersal structure".

P. 2, L. 64: Thomson et al. (2011) found that the higher dispersal distance of large-seeded plants resulted from the fact that these plants were taller compared to the small-seeded species. Authors concluded: "Once plant height was accounted for, we found that small-seeded species dispersed further than did large-seeded species".

P. 2, L. 80-81: This is a shortcut, in my opinion. In endozoochory, maternal large-seeded plant invests more energy in seeds provisioning to make them to be attractive for animals. If something else is possible, explain, please.

P. 3, L. 127-130: If the samples are suitable for statistical analyses, it would be interesting to check such relationships dividing species into those with pappuses, wings and awns.

P. 3, L. 137: This is not significant result at alfa = 0.05.

P. 7, L. 229: The citation is incorrect. The correct form is:

Pyšek, P., Richardson, D.M. (2008). Traits Associated with Invasiveness in Alien Plants: Where Do we Stand?. In: Nentwig, W. (eds) Biological Invasions. Ecological Studies, vol 193. Springer, Berlin, Heidelberg. https://doi.org/10.1007/978-3-540-36920-2_7

P. 7, L. 252: dot is needed between words “variance” and “Nonetheless”

P. 8, L. 307: What kind of relationships were analyzed? Spearman? Pearson?

P. 8, L. 308-310: These are results, and they should be placed in the proper chapter.

P. 8, L. 312: Please, sort the species alphabetically within the groups represented by small and large seeds in Table S2.

P. 9, L. 318-320: Why was 1.5 mg value taken as limit between small and large seeds? What is the justification for this? Did authors try to test another value?

Author Response

Reviewer 2 Comments and Suggestions for Authors

In general, manuscript “Seed size, seed dispersal traits, and plant dispersion patterns for native and introduced grassland plants” (plants-2213896) by Jane E. Tuthill, Yvette K. Ortega, and Dean E. Pearson is very interesting and well written. Authors studied the relationship between size and seed dispersal traits in native and exotic grassland species in western Montana, USA in order to assess factors that influence the efficiency of active dispersal. In the face of biological invasions, I find this this subject to be important. Language is clear and smooth. However, I suggest some corrections and explanations in the text:

  1. 1, L. 32-35: The construction of this sentence is strange. On the one hand, authors wrote: "From a theoretical perspective...", on the other hand, the papers tested this hypothesis were cited. Please, explain the point of hypothesis in the first sentence, and later, give suitable examples.

Response: We have made edits to this sentence to address this point.

  1. 2, L. 60-62: It is not always true. What about the seeds dispersed due to the gravity (Quercus, Fagus, Aesculus, Lagenaria, Cocos)? In so formulated sentence, height of maternal plant (from the next paragraph) cannot be considered as "dispersal structure".

Response: We agree that this pattern is not always true.  However, the sentence states “Global-scale tests of dispersal trade-offs have found that larger-seed sizes were associated with greater dispersal distances overall,” and this statement is true as supported by the following citations which include global scale analyses.  Accordingly, we have not changed that statement.

  1. 2, L. 64: Thomson et al. (2011) found that the higher dispersal distance of large-seeded plants resulted from the fact that these plants were taller compared to the small-seeded species. Authors concluded: "Once plant height was accounted for, we found that small-seeded species dispersed further than did large-seeded species".

Response: Yes, Thomson et al. (2011) conducted a separate analysis wherein they controlled for plant height when examining the seed mass-dispersal question.  We chose to first highlight the pattern they observed when covarying traits such as plant height were not accounted for, which showed a positive relationship between seed mass and dispersal.  We state after the Thomson citation that when Tamme et al. (2014) conducted a similar study that accounted for plant height and considered far more species, this pattern reversed.  To acknowledge that this result aligns with Thomson et al. (2011)’s followup analysis, we have added reference to the latter paper here as well.

  1. 2, L. 80-81: This is a shortcut, in my opinion. In endozoochory, maternal large-seeded plant invests more energy in seeds provisioning to make them to be attractive for animals. If something else is possible, explain, please.

Response:  We agree with the reviewer that at some point, as seed size increases, parental investment in seed size is a means of achieving dispersal by means of endozoochory.  We do not see a conflict with this idea in the referenced statement.  Moreover, this idea is supported by the given references.

  1. 3, L. 127-130: If the samples are suitable for statistical analyses, it would be interesting to check such relationships dividing species into those with pappuses, wings and awns.

Response: We agree with the reviewer, and we tried this analysis early on.  The results suggest that indeed having dispersal structures facilitates greater movement and broader dispersal, but we did not feel that the sample sizes were sufficient for a robust result, so we did not include these results.

  1. 3, L. 137: This is not significant result at alfa = 0.05.

Response: all the p-values referenced on line 137 are P < 0.05, which is generally considered to be significant.

  1. 7, L. 229: The citation is incorrect. The correct form is:

Pyšek, P., Richardson, D.M. (2008). Traits Associated with Invasiveness in Alien Plants: Where Do we Stand?. In: Nentwig, W. (eds) Biological Invasions. Ecological Studies, vol 193. Springer, Berlin, Heidelberg. https://doi.org/10.1007/978-3-540-36920-2_7

Response: We have corrected the citation.  We thank the reviewer for pointing this out.

  1. 7, L. 252: dot is needed between words “variance” and “Nonetheless”

Response: We fixed this typo.  Thank you.

  1. 8, L. 307: What kind of relationships were analyzed? Spearman? Pearson?

Response: We now clarify in the text that Spearman correlation coefficients are presented.

  1. 8, L. 308-310: These are results, and they should be placed in the proper chapter.

Response: In this case, we used simple statistics to justify our methodological approach, which focused analysis on seed mass as our measure of seed size.  For this reason, we believe it is appropriate to retain the material in the Methods section.

  1. 8, L. 312: Please, sort the species alphabetically within the groups represented by small and large seeds in Table S2.

Response: We have retained the present order of the species in this table, which sorts species by seed mass.  This allows the reader to readily see the range of seed sizes across species, particularly the natural break in seed sizes used to define seed size categories.  This seems much more informative for readers than an alphabetical sequence.

  1. 9, L. 318-320: Why was 1.5 mg value taken as limit between small and large seeds? What is the justification for this? Did authors try to test another value?

Response: This was a natural break in the data between large and small seeds, as explained in detail in the Methods and now alluded to in the legend.  This break can be seen in the table and also Figure 1 (note that the break is 1.5 mg or 0.9 mg natural log scale) and is also discussed in our response to Reviewer 1.

Reviewer 3 Report

Lines 229-230: wrong reference Pysek et al. 2007

Author Response

Reviewer 3 Comments and Suggestions for Authors

Lines 229-230: wrong reference Pysek et al. 2007

Response: We have changed the reference here and in another location per the suggestion of this reviewer and also reviewer 2. 
